# Serotype Distribution, Virulence Determinants and Antimicrobial Susceptibility of *Streptococcus agalactiae* Isolated from Young Infants

**DOI:** 10.3390/pathogens11111355

**Published:** 2022-11-15

**Authors:** Zhengjiang Jin, Juan Li, Haijian Zhou, Zhenhui Wang, Lu Yi, Nian Liu, Jiaxi Du, Chien-Yi Chang, Wenjing Ji

**Affiliations:** 1Department of Clinical Laboratory, Maternal and Child Health Hospital of Hubei Province, Wuhan 430070, China; 2State Key Laboratory of Infectious Disease Prevention and Control, National Institute for Communicable Disease Control and Prevention, Chinese Center for Disease Control and Prevention, Beijing 102206, China; 3Department of Pharmacy Administration and Clinical Pharmacy, School of Pharmacy, Xi’an Jiaotong University, Xi’an 710061, China; 4School of Dental Sciences, Faculty of Medical Sciences, Newcastle University, Newcastle upon Tyne NE2 4BW, UK

**Keywords:** Group B Streptococcus, serotype, surface protein, virulence determinants, antibiotic resistance

## Abstract

**Background**: *Streptococcus agalactiae* (Group B Streptococcus, GBS) is the most common cause of serious infections in the first 3 months of life worldwide. The pathogenicity of GBS is closely related to serotypes, surface proteins and virulence factors, and the distribution of them may vary temporally and geographically. However, data related to GBS surface proteins and virulence determinants in China are very few. The aim of this study is to investigate the genetic characteristics of clinical GBS isolates from infected infants. **Methods**: We recovered GBS isolates from infected infants younger than 3 months during 2017–2021 at Maternal and Child Health Hospital of Hubei Province in China. We assessed the GBS serotypes, surface proteins, virulence determinants and antibiotic resistance genes distribution, by Multilocus sequence typing (MLST) and whole-genome sequencing analysis. **Results**: Among 97 isolates (81 EOD and 16 LOD), 5 serotypes were detected. Serotype III was the most represented (49.5%), followed by type Ib (20.6%). The isolates belonged to 17 different sequence types (STs) that grouped into the 8 clonal complexes (CCs). The most frequently identified ST was ST17 (23.7%). The most predominant surface protein of alpha-protein-like (alp) family (one of the protein components of the GBS surface antigen, resistant to trypsin) present was Rib (41.2%), which was mainly detected in serotype III. The *srr1*, which encodes Srr1 protein, was identified in 54.6% of isolates. The *hvgA* encoding for hypervirulent GBS adhesin can be detected in all 24 serotype III GBS. Among the pilus islands genes, 50% and 58.8% of the isolates were positive for *pi-1* and *pi-2a* genes, respectively. The presence of *pi-2b* was mainly associated with serotype III/CC17 strains; 56.7% of isolates carried *tetM*, *tetO/tetL*, *ermB* antibiotic resistant genes. Among all the virulence genes detected, the *cfb-cylE-lmb-pavA* pattern was the main virulence gene profile (81.4%), mainly in serotype III/CC17. **Conclusions**: The whole genomic sequencing data revealed the high variation in surface proteins, determining virulence and antibiotic resistance in clinical isolates from 97 GBS infected infants. These data provide insightful characteristics of genetic features of GBS. Constant epidemiological surveillance is warranted to provide information on the GBS pathogenic dynamics and antibiotic resistance profiles in the surveyed areas for improving therapeutic outcomes.

## 1. Introduction

*Streptococcus agalactiae* (Group B Streptococcus, GBS) is a common Gram-positive bacterial colonizer in the genital and the gastrointestinal tract and around 18% of pregnant women are colonized by GBS worldwide [1]. The vertical transmission from GBS colonized mothers to their newborns during pregnancy and labor can result in severe infections such as sepsis, pneumonia, and meningitis, which are the leading cause of neonatal morbidity and mortality in the worldwide [2]. A global total of 19.7 million pregnant women were estimated to have rectovaginal colonization with GBS in 2020, and 231,800 early-onset (infections occurring within the first 7 days of life) and 162,200 late-onset (occurring between 7 days and 89 days after birth) infant invasive GBS cases were estimated to have occurred [3]. Our previous multi-centered, population-based study across 16 provinces of China showed an overall GBS incidence rate of 0.31 (95% CI 0.27–0.36)/1000 [4], which is similar to estimates from elsewhere in Asia (0.30 (95% CI 0.43–0.56)/1000) [5], but differs from the estimate from a meta-analysis in China including Taiwan, Hong Kong, and Macau (0.55 (95% CI 0.35–0.74)/1000) [6].

To treat and prevent GBS infections in young infants, the use of antibiotics is considered one of the most effective means, but with several concerns. In developed countries, GBS remains the leading cause of neonatal sepsis, and efforts regarding the administration of intrapartum antimicrobial prophylaxis (IAP) have significantly reduced the incidence of neonatal sepsis, especially early-onset GBS infections (EOD) in GBS-colonized pregnant women [7]. In the United States, IAP involving ampicillin administration has been carried out since 1996 to prevent the vertical transmission of GBS. However, IAP has no effect on reducing late-onset GBS infections (LOD), and reduced susceptibility to antibiotics has been reported worldwide, especially the resistance of GBS to macrolides conferred by *ermB*, *ermTR*, and *mefA/E* genes [6,8,9,10]. 

Recent development of the GBS vaccine has raised high hopes. Implementing a vaccine for pregnant women is a promising strategy to prevent neonatal and infant GBS disease and has been identified as a priority by the World Health Organisation (WHO). GBS virulence is complex and multifactorial. A number of virulence factors expressed by GBS are involved in colonization, adherence, invasion and immune evasion and these could be used as potential vaccine candidates [11]. Capsular polysaccharides (CPS) are recognized as playing a key role as virulence factors and used for classifying GBS isolates into ten serotypes: Ia, Ib, II-IX [12]. These are important targets for the development of capsule polysaccharide-based vaccine strategies. GBS strains can also be classified on the basis of surface proteins, including Alp family proteins, serine-rich repeat proteins, and pilus islands (PIs), etc. [13,14]. Proteins such as HvgA, Rib and PI proteins have been associated with invasiveness of GBS strains [15]. Therefore, CPS, and components of ancillary and backbone proteins of pilus, and HvgA were explored as the vaccine candidate’s targets [16].

Currently, several GBS vaccine candidates are in development, such as multivalent GBS bacterial CPS-CRM197 conjugate vaccines and CPS-protein conjugates vaccines [17,18,19]. Considering that the development of GBS vaccines needs to be informed by surface epitopes of GBS from all over the world, the aim of our study is to describe the serotype distribution, presence of virulence determinants and antimicrobial resistance genes among GBS isolates in infants younger than three months of age collected from the Maternal and Child Health Hospital of Hubei Province, a large tertiary hospital in central China. Our data will provide insightful information on surface proteins and antibiotic resistance profiles of clinical GBS isolates in China.

## 2. Methods and Materials

### 2.1. Clinical Isolates

Ninety-seven GBS isolates recovered from infants younger than 3 months at the Maternal and Child Health Hospital of Hubei Province during 2017 to 2021 were included in this study, while duplicate strains from the same patient were excluded. The study hospital is the largest urban tertiary maternal and children hospital, and is a tertiary care centre for women and children in Wuhan city in central China, caring on average for 6000 children hospitalized annually, with adequate research capabilities and laboratory facilities.

### 2.2. Ethical Approvals

As a retrospective study, individual informed consent was waived by the Ethics Committee of the Maternal and Child Health Hospital of Hubei Province, because this study used data previously collected during the course of routine diagnosis and did not pose any additional risks to the patients. The patient records/information were anonymized and deidentified prior to analysis.

### 2.3. Laboratory Methods

Microbial identification and culture were conducted according to routine diagnostic standard operating procedures used in the clinical laboratory of the study hospital: 1–3 mL of blood or cerebrospinal fluid was collected from neonates, and this sample was cultured using an automated BacT/ALERT 3D system or BD BACTEC™ system. Incubation was continued until positive results were observed or for up to 5 days. Positive cultures were subcultured and identified.

After GBS was incubated in a 35–37 °C, 5% CO_2_ incubator for 18–24 h, wet colonies with a diameter of about 3–4 mm appeared on the sheep blood agar plate, which were gray-white translucent, with a narrow β-hemolytic ring. GBS strains were identified by matrix-assisted laser desorption ionization time-of-fight mass spectrometry (MALDI-TOF MS; Bruker Daltonics, Bremen, Germany), following the manufacturer’s instructions. After collection, the strains were cultured on blood agar plates, and the genomic analysis was performed in the State Key Laboratory of Infectious Disease Prevention and Control of the National Institute for Communicable Disease Control and Prevention in the Chinese Center for Disease Control and Prevention.

### 2.4. Whole Genome Sequencing

Genomic DNA was extracted using the Wizard Genomic DNA Purification Kit (Promega, Madison, WI, USA) for genome sequencing. Genomic DNA libraries were prepared using the Nextera XT library preparation kit (Illumina, San Diego, CA, USA) and quantified using the Kapa Biosystems library quantification kit following Illumina instruction. DNA paired-end sequencing was performed by Illumina HiSeq 2000 (Illumina, San Diego, CA, USA). Sequencing reads were de novo assembled using the SPAdes software [20].

### 2.5. Molecular Subtyping

Antibiotic resistance genes, serotypes, surface protein genes and hypervirulent genes were determined according to the GBS bioinformatics pipeline of USA CDC [21]. The ARG-ANNOT and ResFinder databases were incorporated to detect additional resistance determinants [22,23]. Other isolates features, including surface protein genes encoding the hypervirulent GBS adhesin (*hvgA*), serine-rich repeat (*srr*) proteins, pilus islands (*pi-1, pi-1b, pi-2a1, pi-2a2, pi-2a3, pi-2a4, pi-2b, pi-2b2*), and alpha protein family (alpha, *alp2/3, alp1* and *rib*) were extracted from the genomic data. For each sequence query, ≥95% identity was set as the threshold for positive results.

### 2.6. Multilocus Sequence Type

Seven housekeeping genes were used for GBS characterization using the MLST scheme, including *adhP, pheS, atr, glnA, sdhA, glcK,* and *tkt* [24]. GBS can be assigned to a sequence type (ST) based on allelic variation of the seven housekeeping genes and further grouped into clonal complexes (CCs). We compared alleles and STs of all GBS isolates with those in the *S. agalactiae* MLST database (http://pubmlst.org/sagalactiae, accessed on 9 October 2022). STs and CCs were obtained according to the database and the information provided by CCs and Serotype was combined to draw a minimum spanning tree using Phyloviz software [4,25].

## 3. Results

### 3.1. General Characteristics

From January 2017 to December 2021, a total 97 unique isolates were collected from the young infants, and among them, 81 and 16 were from EOD and LOD, respectively. Among them, 56.7% were boys, and the median birth weight was 3150 (interquartile range 2757–3450) grams. Among the prenatal factors, the median gestational age was 39 weeks (interquartile range 37–40), 96.9% were singleton pregnancy, and 61.9% were first pregnancies, 17.5% had premature rupture of membranes, and 63.9% were vaginal births. After birth, the 1-minute and 5-minute Apgar scores were 9 and 10, respectively, and the median length of ICU stay was 11 days (interquartile range 7–16). Among the 97 GBS infection cases, 44.3% were associated with GBS isolates from blood, followed by gastricus (30.9%) and sputum (17.5%). A few strains were collected from cerebrospinal fluid, ear canal secretion and umbilical secretion (Table 1).

### 3.2. Serotype, Sequence Type, and Clonal Complex

A total of 97 available isolates were sequenced. These whole-genome sequencing results have been deposited at DDBJ/ENA/GenBank under the accession JAPFRC000000000 and JAPHEI000000000-JAPHHZ000000000.

A total of 97 available isolates were typed, and 5 of 10 known capsular serotypes were detected (Ia, Ib, II, III and V). The predominant serotype was III (48, 49.5%), followed by Ib (20, 20.6%) and Ia (14, 14.4%). We did not identify isolates of serotypes IV, VI, or others in the study. The proportions of serotypes differed between cases of EOD and LOD (*p* < 0.001) (Figure 1).

**Figure 1 pathogens-11-01355-f001:**
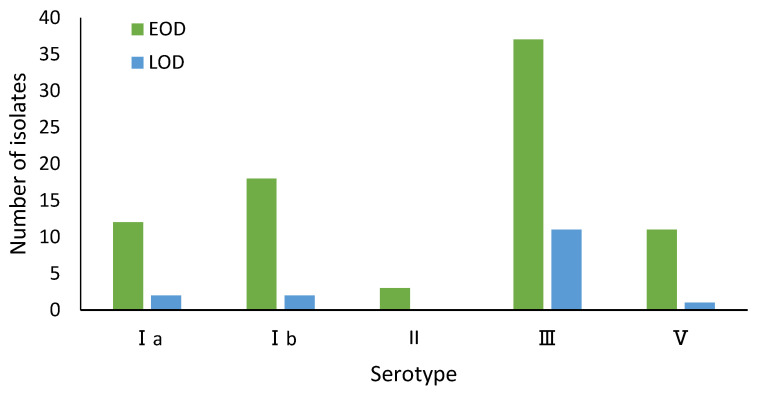
Serotypes of GBS associated with cases of disease onset The isolates belonged to 17 different STs that can be grouped into 8 CCs. The most frequently identified ST were ST17 (23, 23.7%) and ST10 (16, 16.5%), which were found exclusively among serotype III and Ib strains, respectively. The next most common ST was ST19 (14, 14.4%), which was associated with serotype III and V strains. ST23 and ST27 were the most common STs among serotype Ia and III strains, respectively (Figure 2).

**Figure 2 pathogens-11-01355-f002:**
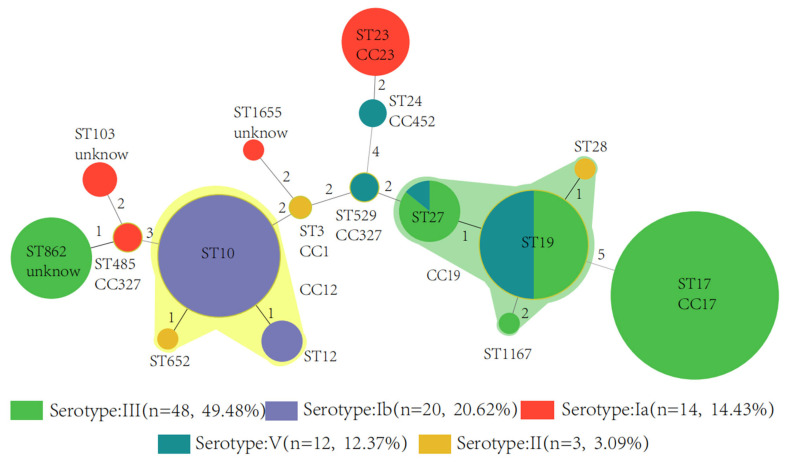
Minimum spanning tree of the 97 GBS isolates by serotype showing the relationship be-tween ST and CC by serotype. Circles represent STs; size of each circle indicates the number of isolates within the specific type. The ST with the greatest number of single-locus variants is the founder ST. GBS isolate serotypes appear as different colors; shading denotes STs belonging to the same CC. The digits 1, 2, 3, 4, 5 are the goeBURST distance scores: given the Hamming distance between profiles. The greater the distance score between different ST types, the farther the genetic distance is. GBS, Group B Streptococcus; CC, clonal complex; ST, sequence type.

### 3.3. Surface Protein Genes

The most predominant alpha-protein-like (alp) family gene present was *rib* in 40 (41.2%) isolates, followed by *alp1* with 31 (32.0%), and 24 (24.7%) isolates were *alpha*. *alp2/3* was not detected in the sample. *rib* (36, 90.0%) was mainly detected in serotype III. *alp1* was detected in Ia (12, 38.7%), III (12, 38.7%) and V (7, 22.6%), while 20 (83.3%) isolates of the *alpha* protein gene were in type Ib. In addition, two isolates were negative for any of the alp family target predictions, and both of them were detected in type Ia (Figure 3).

*pi-1* were detected in 49 (50.0%) and *pi-2a* in 57 (58.8%) isolates. Isolates with serotypes Ia (8), Ib (20), II (3), III (14) and V (12) were *pi-2a* positive, while the pilus subunit query for *pi-2b* was identified in serotype Ia (6) and III (34) isolates in the study. The is *pi-2b* gene was mainly found in serotype III/CC17 strains, while no CC17 strains carried *pi-2a* gene. In addition, almost half (43, 44.3%) of the isolates were positive for *pi-1* and *pi-2a* genes (18 serotype Ib, 13 serotype III, 9 serotype V, and 3 serotype II) (Table 2).

### 3.4. Antimicrobial Resistance Genes

The detection of antibiotic resistance determinants is shown in Table 2. Of the genes which confer resistance to macrolides, *ermB* was detected in 55 (56.7%) isolates, *ermTR* in 9 (9.3%) isolates and *ermT* in 1. The determinant for macrolide efflux, *mef,* was detected in 23 (23.7%) isolates, and lincosamide nucleotidyltransferase determinant, *lnu*, was present in 11 (11.3%). The *ermB* determinant was mainly present in serotypes Ib and III. The only one *ermT* was present in serotype III/CC17. Among 97 isolates, 52 were detected as conferring resistance genes, *tetM*, *tetO* or *tetL*; 26 (78.8%) *tetO* were identified in serotype III. The *Cat* gene, encoding chloramphenicol acetyltransferases, was detected in only one isolate among serotypes II.

### 3.5. Virulence Genes

As shown in Figure 4, the *cylE* and *pavA* gene were detected in all sample isolates, and the *cfb* and *lmb* gene were identified in 99.0% (96/97) and 81.4% (79/97), respectively. The prevalence of the *hylB* virulence genes was 49.5%, and *fbsA* gene was carried by only one isolate. The *cfb-cylE-lmb-pavA* pattern was the main virulence gene profile (79, 81.4%), followed by the *cfb-cylE-lmb-pavA-hylB* pattern (46, 47.4%). The isolates carrying these virulence genes, except *fbsA,* were mainly serotype III/CC17.

## 4. Discussion

By studying the virulence and drug resistance of GBS, we can understand the pathogenicity and drug resistance mechanism of this pathogen [26]. The surface proteins profiles can further provide critical information for the development of vaccines to protect newborn infants against GBS infections. We reported the genetic characteristics of 97 GBS isolates recovered at a large tertiary hospital in Hubei, China, from neonates younger than 3 months over a period of 5 years. We investigated serotype distribution, surface proteins, virulence genes and resistance determinants using the whole genome sequencing analysis. Only a few surveillance studies on GBS infections have been conducted in China, and data on the genetic characteristics of the prevalent circulating lineages are still limited. This study will help form a portrait of the current landscape of GBS pathogenicity and resistance in China.

Most of the 97 cases of GBS infection collected in this study were from children with EOD infection. It is worth noting that 17.5% of the infected patients developed premature rupture of membranes. GBS has strong penetrating capacity, and the pathogenic bacteria adsorbed on the rectum and urethra will directly infect the fetal membrane, and directly invade through the proteolytic enzyme produced by the bacteria itself and the cytokines produced after the phagocytosis of inflammatory cells, which will cause local fetal membrane damage resulting in reduced tension [27]. Furthermore, once inflammation occurs, the fetal membranes of pregnant women will experience edema and degeneration. Under the action of metabolites, premature rupture of membranes will occur after the tension of fetal membranes is reduced. Previous study has shown that the GBS positive rate of pregnant women with premature rupture of membranes is about 20%, similar to this study [28].

GBS possesses multiple virulence factors, among which capsular polysaccharide is an important virulence determinant for invasive infection (Lemire et al., 2012). The type of capsular polysaccharide plays an important role in epidemiological research. The anti-capsular polysaccharide antibody also has protective immunity and is an important virulence factor against phagocytic function. Therefore, study of the capsular polysaccharide of GBS has gained great attention. The early GBS vaccine was also made of the capsular polysaccharide expressed at a high level on the surface of GBS as the target antigen; the tolerance was good and the immunogenicity was poor [29]. The fetal and neonatal protective effect is not obvious, which limits its clinical application [30,31]. Capsular polysaccharides are the main components in the outer capsule of the GBS cell wall. Among them, four (types Ia, Ib, II, and III) are the main strain types, and type III is the main pathogen causing serious infections [32]. In this study, five capsular serotypes (Ia, Ib, II, III and V) were detected, half of which were type III. Serotypes Ia, Ib, II, III, and V were most frequently associated with invasive infections, with a higher proportion of EOD than LOD in serotype III-infected cases (*p* < 0.001). This is inconsistent with reports that GBS of serotype III is commonly seen in neonatal late-onset infection cases. The reason was possibly due to the differences in the prevalence and distribution of GBS serotypes, geographic location and ethnicity. Among different serotype strains, the virulence proteins and their clinical manifestations are also different [33,34]. It has been reported that in neonates, the highly toxic clonal complex 17 (CC17) of serotype III is responsible for GBS meningitis. In this study, 97 GBS isolates belonged to 17 different ST types, and these ST types were divided into 8 CCs, of which ST17/CC17 was more common in infantile diseases, and most ST17/CC17 isolates were serotype III, consistent with the report [35].

The leading GBS vaccine candidates under development are capsular polysaccharide-protein conjugate vaccines [36]. The capsular polysaccharide conjugate vaccine is a combination of capsular polysaccharide and protein to induce capsular-specific antibody response. It is well tolerated and has stronger functional activity and immunogenicity than capsular polysaccharide vaccine, surface protein vaccine and fimbriae vaccine. The immunity to GBS capsular polysaccharide triggered by vaccine can last for at least 18–24 months, which means pregnant women have functionally active antibodies throughout pregnancy after vaccination, which could theoretically prevent invasive infant infections [37]. Five capsular serotypes (Ia, Ib, II, III, and V) were detected in this study; therefore, our results suggest that a pentavalent conjugate vaccine (containing serotypes Ia, Ib, II, III, V) can cover 100% of small infantile pathogenic isolates.

Other major virulence factors include the following cell surface proteins that mediate adhesion and invasion: Alpha-like protein (Alp) family (Alpa, Alp2/3, Alp1, Rib), laminin binding protein (Lmb), fibrin binding surface proteins (FbsA), GBS adhesion proteins (HvgA), serine-rich repeat glycoproteins (Srr1 and Srr2), and pilus islands (PI-1, PI-1b, PI-2a1, PI-2a2, PI-2a3, PI-2a4, PI-2b, PI-2b2) [38,39]. Studies have shown that these proteins are closely related to GBS adhesion to the host, invasion of mucosal epithelial cells, promotion of platelet aggregation, and anti-phagocytic phagocytosis. There is also production of various enzymes/toxins, such as β-hemolysin/cytolysin (*cylE*), hyaluronidase (*hylB*), adherence and virulence protein A (*pavA*), and CAMP factor pore-forming toxin (*cfb*), which facilitate bacterial entry into the organism and survival [40,41].

This study echoes similar findings from another group in China [42]. Rib (41.2%) was the most common of all the alpha-like proteins among the isolates in this study, followed by Alp1 (32.0%). Other studies from have documented Alp1 as the most frequently detected Alp gene in other countries such as Nigeria [43] and Egypt [44]. As protein antigen-based vaccines could provide an alternative to the multivalent polysaccharide-protein conjugate GBS vaccines [45], we calculated the percentage of isolates with at least one of the four Alp family surface protein targets (Alpha, Alp1, Alp2/3 and rib). Based on our results, a protein-alum adjuvant vaccine containing the alp family surface proteins will have the potential to prevent up to 97.9% of infant GBS disease. In addition to the Alp protein-based vaccines, the pilus-like structure plays an important role in GBS pathogenicity, making it another potential target for GBS vaccine development [46]. We found that almost half of the isolates were positive for PI-1 and PI-2a genes. Our study confirmed that the PI-2b is associated with the high pathogenicity of CC17 strains, consistent with the findings from a previous study [47]. The PI-2b enhances GBS adhesion and the tolerance to host phagocytosis.

In recent years, studies have found that the CC17GBS strain of serotype III can exhibit more affinity for nerve cells (neurotropic) than other CCs GBS strains by expressing the sHIP-attaching factor HvgA. HvgA can help GBS to adhere effectively and crosses the gut epithelial cells into the bloodstream as the blood circulates across the blood-brain barrier and invades the brain. Some studies suggest that the HvgA protein can serve as a unique biomarker for HvgA^+^ pathogenic GBS causing severe neuroinvasive GBS infectious diseases [48]. In this study, ST17 was more common in infant disease, and most ST17 isolates are serotype III. ST17 strains normally express HvgA, a virulence factor associated with adhesion and penetration [49]. Our results confirmed that all 24 *hvgA* positive GBS isolates were only found in serotype III/CC17 isolates, consistent with the report [48].

In terms of another surface adhesin, the serine-rich repeat protein, Srr-1, was identified in 54.6% of isolates in this study. In addition, 95.8% of Srr2 was associated with CC17, known to be associated with greater binding affinity, higher morbidity and leading to the development of meningitis [50,51].

In the study of virulence genes, we also found the serotype III/CC-17 GBS strain with a high positive rate of virulence determinants that cause GBS neonatal infections. *cylE* and *pavA* genes were detected in all isolates, and 99.0% of the sample carry the *cfb* gene. These genes belong to the virulence determinants and play important roles in the pathogenesis of illnesses caused by GBS, and they are also acknowledged as vaccine candidates [52]. Furthermore, the *cfb-cylE-lmb-pavA* pattern was the most dominant virulence gene profile (81.4%) in this study, and was different from what was reported in other regions and countries [53]. Our results suggest that vaccines should contain these virulence genes as good targets due to their frequent occurrence.

Antibiotic resistance of GBS is an increasing threat to limit treatment options. Penicillin and ampicillin are the drugs of choice for prevention or treatment of GBS infections, and the sensitivity rate has remained at a very high level [54]. In this study, we analyzed the resistance genes of GBS to other several antibiotics, including erythromycin/clindamycin resistance genes (*ermB, mef, lnuB, ermT, ermTR*); tetracycline resistance genes (*tetM, tetO, tetL*); active efflux pump genes *(lsaC, lsaE*), and the chloramphenicol resistance gene (*cat*).

Macrolide antibiotics have always been important antibiotics for the treatment of community-acquired pneumonia, and macrolides and related drugs are the recommended alternatives for GBS infectious patients who are allergic to β-lactam agents [55]. Thus, the widespread use of erythromycin and other macrolide antibiotics may increase the risk of erythromycin-resistant organisms emerging [56]. It is worth noting that tetracycline, as a broad-spectrum antibiotic, is not widely used in the treatment of streptococcal infection, but the detection rate of tetracycline resistance genes in GBS strains is also high. We found in this study that at least one of the tetracycline resistance determinant genes (*tetM, tetO and tetL*) was present in 53.6% of the sample isolates, and 56.7% of isolates carried *ermB*. An association between antibiotic resistances and certain serotypes or STs has been identified [57]. The majority of our isolates carrying *ermB* belonged to CC-12/17/19. The resistance of GBS to macrolides has been increasing worldwide [58,59], and previous studies have proved that the *ermB*, *ermTR*, and *mefA/E* genes are involved in the resistance to macrolides [60]. The high level of antimicrobial resistance will definitely guide empirical antibiotic therapy to prevent the development of GBS infections and affect the guidelines of GBS prevention and treatment. Thus, continuous monitoring of antimicrobial susceptibility profiles is needed. We did not detect *van* series genes related to vancomycin resistance in this study. The *cat* gene, encoding chloramphenicol acetyltransferases, was detected in only one isolate among serotypes II.

## 5. Conclusions

This study evaluated the features such as virulence factors, surface protein, and resistance genes of GBS isolates by using whole genome sequencing strategy. The findings obtained from this study shed light on the need for a more rigorous characterization and detection of correlations among serotypes, clonal clusters and resistance genes of GBS isolates circulating in the study areas. It will provide important insights into the pathogenic mechanism of GBS infection among infants and for vaccine development in the era of IAP. Our findings also show the need for ongoing surveillance studies in the future.

## Figures and Tables

**Figure 3 pathogens-11-01355-f003:**
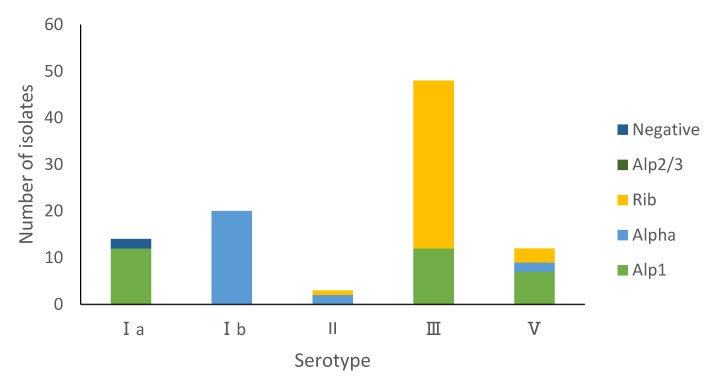
Distribution of GBS alpha family protein types among isolates. Serotype serine-rich repeat proteins gene *srr1* was identified in 53 (54.6%) isolates and *srr2* in 24 (24.7%), while 20 (20.6%) isolates were negative for both. *srr2* was associated primarily with CC17 (23, 95.8%). Isolates negative for *srr* genes were mostly of CC19 (12, 60.0%). All 24 *hvgA* positive GBS isolates were serotype III (CC17, 23; unknown, 1) (Table 2).

**Figure 4 pathogens-11-01355-f004:**
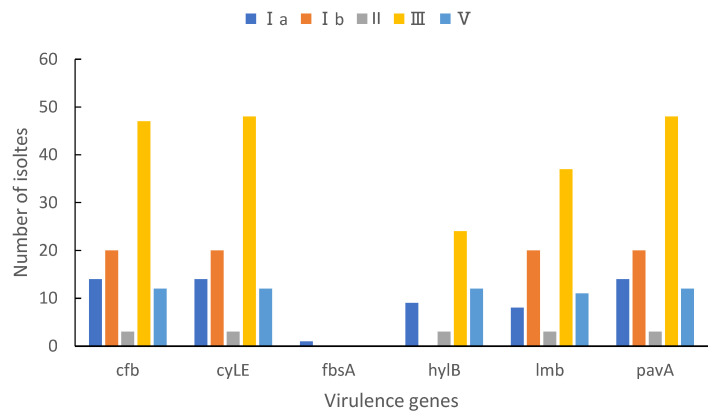
Virulence genes’ associations with GBS serotypes.

**Table 1 pathogens-11-01355-t001:** Characteristics of *Group B Streptococcus* isolates from infants aged younger than 3 months, by disease onset.

	EOD (n = 81)	LOD (n= 16)	Total (n = 97)
Age (days)			
Median (IQR)	1 (1–1)	20 (17–24)	1 (1–1)
Gender (n, %)			
Male	47 (58.0)	8 (50.0)	55 (56.7)
Female	34 (42.0)	8 (50.0)	42 (43.3)
Birth weight (grams)			
All cases (median (IQR))	3150 (2600–3400)	3200 (3000–3515)	3150 (2757–3450)
Apgar-1min			
Median (IQR)	9 (9–10)	10 (8–10)	9 (9–10)
Apgar-5 min			
Median (IQR)	10 (10–10)	10 (10–10)	10 (10–10)
Single or twins (n, %)			
Single	79 (97.5)	15 (93.8)	94 (96.9)
Twins or multiple	2 (2.5)	1 (6.3)	3 (3.1)
Gestational age (weeks)			
Median (IQR)	39 (36–40)	39 (38–40)	39 (37–40)
Delivery mode (n, %)			
Vaginal	53 (65.4)	9 (56.3)	62 (63.9)
C-section	28 (34.6)	7 (43.8)	35 (36.1)
Specimen source (n, %)			
Blood	31 (38.3)	12 (75.0)	43 (44.3)
Gastricus	30 (37.0)	0 (0.0)	30 (30.9)
Sputum	17 (21.0)	0 (0.0)	17 (17.5)
Cerebrospinal fluid	1 (1.2)	4 (25.0)	5 (5.2)
Ear canal secretion	1 (1.2)	0 (0.0)	1 (1.0)
Umbilical secretion	1 (1.2)	0 (0.0)	1 (1.0)
Premature rupture of membranes (n, %)			
No	65 (80.2)	15 (93.8)	80 (82.5)
Yes	16 (19.8)	1 (6.3)	17 (17.5)
Pregnancy type (n, %)			
First time	52 (64.2)	8 (50.0)	60 (61.9)
Non-first	29 (35.8)	8 (50.0)	37 (38.1)
Length of stay in ICU (days)			
Median (IQR)	10 (7–15)	14 (8–30)	11 (7–16)

EOD, early-onset-disease; LOD, late-onset-disease; SD, standard deviation; IQR, interquartile range; n, number.

**Table 2 pathogens-11-01355-t002:** The characteristics of GBS by serotype, resistance determinants and other strain characteristics.

Serotype	Srr	Pili	HvgA	Macrolide/ Clindamycin Genes	Tetracycline Resistance Genes	LsaE/LsaC	Chloramphenicol Genes
Ia	*srr1* (11)	*pi-1* (1); *pi-2a* (8); *pi-2b* (6); *pi-1*:*pi-2b* (1)		*ermB* (1); *mef*(4); *lnuB* (1)	*tetM* (3); *tetO* (1); *tetL* (1)	*lsaC* (3); *lsaE* (1)	
Ib	*srr1* (17)	*pi-1* (18); *pi-2a* (20); *pi-1:pi-2a* (18)		*ermB* (20); *lnuB* (1)	*tetO* (3)	*lsaE* (1)	
II	*srr1* (3)	*pi-1* (3); *pi-2a* (3); *pi-1:pi-2a* (3)		*ermB* (1); *ermTR*(4)	*tetM* (2); *tetL* (1)		*cat* (1)
III	*srr1* (14); *srr2* (24)	*pi-1* (18); *pi-2a* (14); *pi-2b* (34); *pi-1:pi-2a* (13); *pi-1:pi-2b* (5)	HvgA (24)	*ermB* (31); *ermT* (1); *mef*(13); *lnuB* (8)	*tetM* (7); *tetO* (26); *tetL* (1);	*lsaC* (1); *lsaE* (8)	
V		*pi-1* (9); *pi-2a* (12); *pi-1:pi-2a* (9)		*ermB* (2); *ermTR*(8); *mef*(6); *lnuB* (1)	*tetM* (7); *tetO* (3); *tetL* (1);	*lsaE* (1)	

## Data Availability

The data presented in this study are available on request from the corresponding author. The data are not publicly available due to privacy restrictions.

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
