# Peer review of "Serotype Distribution, Virulence Determinants and Antimicrobial Susceptibility of Streptococcus agalactiae Isolated from Young Infants"

_pathogens, 2022, doi:10.3390/pathogens11111355_

Round 1
Reviewer 1 Report
This manuscript uses genomic sequencing to predict the serotype, virulence determinants and antimicrobial susceptibility of Group B streptococcus isolated from 97 young infants in a hospital in China. Although limited to a single geographic area the study is well justified and well presented. However, there is a lack of technical information that should be easily addressed. Section 2.4 should be expanded to directly associate GBS genomic preparation, library construction and whole genome sequencing with their respective product manufacturers. The library kit, Illumina instrument and length of the reads produced (for example, 150 bp, paired end reads) need to be specified. In addition, the method of assembling the reads into contigs needs to be stated. There are numerous grammatical errors, I have only noted the ones that affect scientific clarity.
Specific comments:
Line 130: How were these genes extracted from the genomic data?
Line 135: Include a reference for the MLST scheme.
Line 140: The reference cited (19) does not mention MLST 2.22.0 software.
Line 151: This statement implies that all infants had Apgar scores of 9 and 10. That seems unlikely.
Line 165: Figure 1. “Isolates” is misspelled on the Y axis label.
Line 178: Throughout most of the manuscript care is taken to specify that genes are being used to predict gene products. However, in section 3.3, the impression is given that proteins were studied directly. This needs to be clarified.
Line 185: Figure 3. “Isolates” is misspelled on the Y axis label.
Line 219: The first sentence of the discussion needs editing for clarity.
Line 234: This sentence implies that inflammatory cells are being phagocytized when the authors likely meant GBS are being phagocytized.
Line 268: Stronger than what? A comparison is called for.
Line 272: The strong conclusion, from a limited study, that 5 serotypes can cover 100% of pathogenic isolates should toned down.
Line 333: Remove one “widespread”
Line 343: A reference is needed
Line 347: Unless vancomycin was tested for phenotypically this should be changed to reflect genetic analysis.
Reviewer 2 Report
Thank you for your manuscript, which presents data from Hubei province on GBS virulence factors and serotypes.
I have a few comments/queries:
1. Introduction - your vaccine references are not from vaccine trials, for example the Madhi study is a natural immunity study. There are good summary papers e.g. Vekemans et al in Vaccine that would be more appropriate.
2. Methods: definitions of culture positivity, sites and collection methods are missing but would be useful to understand e.g. meningitis vs blood stream infection
Are the genes in a publically available database for comparison with other populations?
What software was used to determine the WGS?
It would be useful to do also phenotypic analysis of the isolates and their antibiotic determinants to see if there is also phenotypic resistance
Reviewer 3 Report
The manuscript brings a relevant theme and has important data for the area.
Minor corrections are indicated in the attached file.
In addition to the small corrections indicated, we suggest the inclusion of a better minimum spanning tree figure.
Considering that 97 strains were sequenced, we believe that more complete analyzes could be performed, in addition to comparing the strains with isolates from other countries.

Round 2
Reviewer 2 Report
comments addressed